# Recent Clinical Isolates of Enterovirus D68 Have Increased Replication and Induce Enhanced Epithelial Immune Response Compared to the Prototype Fermon Strain

**DOI:** 10.3390/v15061291

**Published:** 2023-05-31

**Authors:** Mark K. Devries, Yury A. Bochkov, Michael D. Evans, James E. Gern, Daniel J. Jackson

**Affiliations:** 1Department of Pediatrics, School of Medicine and Public Health, University of Wisconsin, Madison, WI 53792, USA; mkdevries@wisc.edu (M.K.D.); yab@medicine.wisc.edu (Y.A.B.); gern@medicine.wisc.edu (J.E.G.); 2Department of Biostatistics, School of Medicine and Public Health, University of Wisconsin, Madison, WI 53792, USA; evan0262@umn.edu; 3Department of Medicine, School of Medicine and Public Health, University of Wisconsin, Madison, WI 53792, USA

**Keywords:** enterovirus D68, respiratory illness, viral replication, RNA-seq

## Abstract

In 2014, enterovirus D68 (EV-D68), previously associated primarily with mild respiratory illness, caused a large outbreak of severe respiratory illness and, in rare instances, paralysis. We compared the viral binding and replication of eight recent EV-D68 clinical isolates collected both before and during the 2014 outbreak and the prototype Fermon strain from 1962 in cultured HeLa cells and differentiated human primary bronchial epithelial cells (BEC) to understand the possible reasons for the change in virus pathogenicity. We selected pairs of closely related isolates from the same phylogenetic clade that were associated with severe vs. asymptomatic infections. We found no significant differences in binding or replication in HeLa cell cultures between the recent clinical isolates. However, in HeLa cells, Fermon had significantly greater binding (2–3 logs) and virus progeny yields (2–4 logs) but a similar level of replication (1.5–2 log increase in viral RNA from 2 h to 24 h post infection) compared to recent isolates. In differentiated BECs, Fermon and the recent EV-D68 isolates had similar levels of binding; however, the recent isolates produced 1.5–2-log higher virus progeny yields than Fermon due to increased replication. Interestingly, no significant differences in replication were identified between the pairs of genetically close recent EV-D68 clinical isolates despite the observed differences in associated disease severity. We then utilized RNA-seq to define the transcriptional responses in BECs infected with four recent EV-D68 isolates, representing major phylogenetic clades, and the Fermon strain. All the tested clinical isolates induced similar responses in BECs; however, numerous upregulated genes in antiviral and pro-inflammatory response pathways were identified when comparing the response to clinical isolates versus Fermon. These results indicate that the recent emergence in severe EV-D68 cases could be explained by an increased replication efficiency and enhanced inflammatory response induced by newly emerged clinical isolates; however, host factors are likely the main determinants of illness severity.

## 1. Introduction

EV-D68, a member of the genus Enterovirus, family Picornaviridae, was first identified in 1962 in a child with pneumonia-like symptoms [1]. EV-D68 infections can cause respiratory illnesses ranging from asymptomatic to upper respiratory symptoms to more severe illnesses including bronchitis, wheezing, and/or shortness of breath. In rare instances, EV-D68 infection has also been associated with polio-like neurological symptoms [2]. Its severity has been further demonstrated in multiple epidemiological studies [3,4] with higher rates detected in a hospital compared to outpatient settings. In 2014, reports of an upper respiratory tract infection that was associated with severe outcomes including hospitalization for respiratory distress and paralysis emerged in the United States. The 2014 outbreak is the largest and most widespread outbreak recognized to date, resulting in significant morbidity and mortality [5], which created a need to understand the mechanisms underlying enhanced EV-D68 pathogenicity.

What makes EV-D68 unique from other enteroviruses are some biologic characteristics in common with rhinoviruses (RV), such as its optimal growth at 33 °C and acid lability. In fact, this similarity of EV-D68 with RV led to it being incorrectly classified as an RV in the past [6,7]. EV-D68 can infect the upper respiratory tract epithelium potentially through multiple different receptors [8,9,10]. However, the physiological relevance of some of these receptors has been questioned [10,11].

While EV-D68 was identified 50 years ago, it has rarely been associated with severe illnesses, with only a total of 26 reported cases from 1970 to 2005, with 75% of them occurring in children [12]. In the late 2000s, the prevalence of EV-D68 infections began to increase [3,4]. In recent years, EV-D68 isolates have been grouped into three major phylogenetic clades designated A, B, and C [13]. Multiple clades have circulated with each outbreak of EV-D68 illnesses, including a novel sub-clade (B1) that emerged in the 2014 outbreak. This trend of EV-D68 evolution has continued in 2019–2020, where multiple clades co-circulated and a novel B3 clade emerged [14]. This has led to speculation that the increased prevalence of EV-D68 relates to mutations that altered its antigenicity [4]. This change in antigenicity could sidestep antiviral immunity, leading to increased EV-D68 infection rates.

The innate immune response is critical to controlling EV-D68 and other respiratory viral infections. An impaired innate immune response can increase the risk of severe illness. Both epithelial and mononuclear cells contribute to innate antiviral responses, which include the secretion of interferons (IFN) and pro-inflammatory cytokines, such as IL-6 and TNF. EV-D68 can inhibit the antiviral response by preventing type I interferon activation in HeLa cells [15], but there is less evidence of IFN inhibition in primary epithelial cells.

We hypothesized that recently emerged isolates of EV-D68 might be more virulent because of enhanced binding and/or replication within epithelial cells or induced unique patterns of gene expression. To test this hypothesis, we compared the properties of eight recent EV-D68 isolates to those of the Fermon strain, which was isolated in 1962, in differentiated cultures of human primary bronchial epithelial cells (BEC).

## 2. Materials and Methods

### 2.1. Virus Isolate Selection

The EV-D68 clinical isolates selected for this study were identified by RT-PCR and partial sequencing in nasal samples from children enrolled in the Childhood Origins of Asthma (COAST) study, the Urban Environment and Childhood Asthma (URECA) study, and the Preventative Omalizumab or Step-up Therapy for Fall Exacerbations (PROSE) study [16,17,18]. The EV-D68 clinical isolates were then fully sequenced and grouped into 3 major clades by phylogenetic analysis of the full-length genomic sequences (Figure 1). Associated symptom data on illness severity were used to select viruses based on the severity of symptoms. We classified EV-D68 strains into those that caused a severe wheezing episode or asthma exacerbation or asymptomatic infections. We selected pairs of closely related viruses from the same clade that were associated with severe vs. asymptomatic infections. Based upon these criteria, we selected 8 different EV-D68 isolates that belonged to 3 different viral clades: A, B, and C (Figure 1). Within clade B, we included two isolates (C7787 and C7788) from the B1 subclade, which was responsible for many severe EV-D68 cases in the 2014 outbreak [19].

### 2.2. Construction of the Full-Length EV-D68 cDNA Infectious Clones

We generated cDNA infectious clones of 8 EV-D68 clinical isolates using the published sequences (Table 1). To construct the full-length cDNA copy of the viral genome, we used 6 overlapping cDNA fragments (gBlocks) for each isolate made by Integrated DNA Technologies (Coralville, IA, USA). The viral genome of each isolate was preceded by the hammerhead ribozyme sequence [20] and followed by the poly(A) tail. We cloned complete viral genomes in two steps using a vector backbone from pR16.11 plasmid [21]. First, we assembled (Gibson Assembly Master Mix, NEB) two large cDNAs corresponding to 5′ and 3′ halves of the viral genome from 3 overlapping gBlocks each. We then linked these two cDNAs by restriction digestion (Table 1) followed by ligation (T4 DNA ligase, NEB) to create a full-length cDNA copy of the viral genome. We also used two Fermon strains, one that was provided to us by Dr. Frank van Kuppeveld (Utrecht University), which we referred to as “Fermon”, and another one that was made using the reverse genetics methods described above and designated as “Lab Fermon”. We sequenced the complete genome of the “Fermon” strain and found three non-synonymous mutations in non-structural proteins, R^1179^Q and T^1410^S in 2C, and I^2009^V in 3D, compared to “Lab Fermon” that was based on published sequence AY426531. For all the experiments comparing Fermon to clinical isolates, we used the recombinant virus produced from a “Fermon” infectious clone.

### 2.3. Recombinant Virus Production in WisL Cells

RNA transcripts were synthesized from the linearized plasmids (Table 1) with a RiboMax large-scale RNA production system T7 (Promega) for 3 h. The reaction products were then treated with RQ1 DNase I (Promega) and analyzed by agarose gel electrophoresis. We used 50 μL of RNA directly from this reaction with 50 μL of Lipofectamine 2000 (Invitrogen) for transfection of two T-75 flasks of WisL cells (human embryonic lung fibroblasts). We diluted 50 μL of Lipofectamine in 4 mL of Opti-MEM (Invitrogen) and incubated it for 5 min. Next, we added 50 μL of viral RNA to the diluted liposomes and then gently mixed and incubated it for 20 min at room temperature. Then, two T-75 flasks of WisL cells were washed with PBS, and 2 mL Opti-MEM was added to the cells. Finally, 2 mL of the RNA–Lipofectamine mixture was added and incubated for 2 h at 34 °C. After 2 h, the transfection medium was replaced with complete cell culture medium (10 mL per flask), and the cells were incubated at 34 °C for 24 h. After 24 h, the transfected cells were collected and stored at −80 °C until purification.

### 2.4. Virus Purification

Transfected WisL cells were frozen and thawed 3 times and scraped. The cell lysate was centrifuged for 10 min at 10,000× *g* and 4 °C to remove cell debris. The clarified lysate was then treated with RNase A (10 μg/mL; Qiagen) for 10 min at 37 °C to remove free RNA. Next, 1 mL of 10% N-lauroylsarcosine and 20 µL β-mercaptoethanol were added per 10 mL of clarified lysate, and ~11 mL of this mixture was layered over 1 mL of 30% (*w*/*v*) sucrose (in PBS) in Ultra-Clear (14 × 89 mm, Beckman) centrifuge tubes. The samples were then centrifuged at 24,200 rpm (SW41 rotor, Beckman L-60) and 10 °C for 4 h.

After 4 h, the supernatant and sucrose were aspirated, 100 µL of cold PBS with 0.01% BSA was added, and the tube was incubated overnight at 4 °C. The virus pellet was then resuspended by pipetting multiple times. Virus suspension was collected and centrifuged at 10,000× *g* for 3 min to remove any remaining debris. Total RNA was extracted from 2 μL of purified viral prep with an RNeasy Mini Kit (Qiagen) as per the manufacturer’s instructions. Next, EV-D68 RNA concentrations were determined by RT-qPCR with a standard curve from 10^2^–10^7^ plaque-forming unit equivalents (PFUe).

### 2.5. Reverse Transcription

A TaqMan Reverse Transcription Reagents kit (Life Technologies) was used for reverse transcription (RT) of viral RNA with virus-specific reverse primer R848 (5′-AAACACGGACACCCAAAGTAGT-3′). The RT master mix was made as per the manufacturer’s recommendation. A total of 3.85 µL of RNA with 6.15 µL of reaction mixture was used, and reverse transcription was performed with a thermal profile of 25 °C for 10 min, 48 °C for 30 min, 95 °C for 8 min, and 4 °C. After this, 40 µL of RNase-free water was added, and the viral RNA concentration was assessed by qPCR.

### 2.6. Quantitative PCR

Quantitative (q) PCR was performed using a Power SYBR green PCR master mix (Applied Biosystems) and 7300 real-time PCR system (Applied Biosystems). The forward (5′-CCTCCGGCCCCTGAAT-3′) and reverse (5′-AAACACGGACACCCAAAGTAGT-3′) primers were used in a 25 µL reaction in duplicate. The thermal profile used was 50 °C for 3 min, 95 °C for 5 min followed by 40 cycles at 95 °C for 15 s and at 60 °C for 30 s. The levels of EV-D68 RNA were determined by a standard curve from serial 10-fold dilutions of virus with a known titer (10^2^–10^7^ PFUe).

### 2.7. Differentiated BEC Cultures Grown at an Air–Liquid Interface

BECs were obtained from de-identified donors. Frozen stocks of BECs were thawed and washed with PneumaCult-Ex medium (Stemcell Technologies). Then, cells were seeded in T-75 CellBind flasks (Costar; Corning Inc., Corning, NY, USA) and cultured in PneumaCult-Ex medium. At 90% confluence, the BECs were washed with PBS and treated with 0.25% Trypsin-EDTA to dislodge cells. The cells were counted and seeded (10^5^ cells per well) in Transwell inserts (0.4 µM pore size, Costar) coated in 5% human placental collagen type VI solution and placed in 24-well plates. The cells were grown in a PneumaCult-Ex medium with antimicrobials (50 mg/mL gentamicin and 2 mg fluconazole) overnight or until confluence at 37 °C. Then, the PneumaCult-Ex medium was replaced with PneumaCult-ALI maintenance medium in the basal chamber. The BEC cultures were grown at an air–liquid interface (ALI) at 37 °C until fully differentiated (approximately 4 weeks).

### 2.8. Infection of BEC-ALI Cultures

Prior to viral infection, the complete growth medium was removed from the BEC-ALI cultures and replaced with a PneumaCult-ALI maintenance medium without hydrocortisone for 24 h. After 24 h, the BECs were infected with 10^5^ plaque-forming unit equivalents (PFUe) per well of purified EV-D68 (two wells per each isolate), which was diluted in 50 µL of PneumaCult-ALI maintenance medium without hydrocortisone. The cultures were gently shaken for 15 min at room temperature and then incubated at 34 °C for 2 h. After 2 h, the cells were washed apically 3 times with PBS, and then 350 μL of the RLT lysis buffer was added to one well (to measure virus binding at 2 h p.i.) and 500 µL of ALI growth medium was added basally to the second well. At 24 h p.i., 350 μL of the RLT lysis buffer was added to cells, and then total RNA was extracted with an RNeasy mini kit (Qiagen) as previously described [22] to measure the virus progeny yield and replication.

### 2.9. HeLa Cell Infections

HeLa cells were grown in T-75 flasks until confluence. The cells were then trypsinized (3 min at 37 °C), spun (500× *g*, 5 min), resuspended in Medium A (EMEM (Lonza 12-611F) supplemented with non-essential amino acids (Gibco 11140), penicillin/streptomycin (Gibco 15140), and 10% fetal bovine serum (Gemini 100-500)) and plated in 12-well plates at a density of 5 × 10^5^ cells per well. After overnight incubation, the cells were infected with different isolates of EV-D68 (2 wells each) at 2.5 × 10^6^ PFUe per well. The viruses were allowed to bind for 2 h and then washed 3 times with PBS. After the wash step, 350 µL of the RLT buffer was added to one well of cells (to measure virus binding at 2 h), and fresh Medium A was added to the second well (to measure virus progeny yield and replication at 24 h). After 24 h, the growth medium was removed, and 350 µL of the RLT buffer was added to the cells. Total RNA was extracted with an RNeasy mini kit (Qiagen) as per the manufacturer’s instructions.

### 2.10. RNA-Sequencing Library Construction and Sequencing of Directional Libraries

BEC cultures from 5 donors were differentiated at ALI. We selected one isolate from each EV-D68 phylogenetic clade: O541a (clade A), C7731 (clade B), C7788 (clade B1), C2386 (clade C), and the Fermon strain. The total RNA samples from the BEC cultures collected 24 h p.i. were submitted to the University of Wisconsin-Madison Biotechnology Center and verified for purity and integrity via a NanoDrop One Spectrophotometer and Agilent 2100 BioAnalyzer, respectively. Then, using Illumina^®^ TruSeq^®^ Stranded mRNA Sample Preparation kits (Illumina Inc., San Diego, CA, USA), each library was prepared and standardized according to Illumina Inc. protocol. For each library preparation, mRNA was purified from 200 ng of total RNA using poly-T oligo-attached magnetic beads. Subsequently, each poly-A-enriched sample was fragmented using divalent cations under elevated temperature. The RNA was synthesized into double-stranded cDNA using SuperScript II Reverse Transcriptase (Invitrogen, Carlsbad, CA, USA) and random primers for first strand cDNA synthesis followed by second strand synthesis. Double-stranded cDNA was purified by paramagnetic beads (Agencourt AMPure XP beads, Beckman Coulter). The cDNA products were incubated with Klenow DNA Polymerase to add an ‘A’ base (Adenine) to the 3′ end of the blunt DNA fragments. DNA fragments were ligated to Illumina adapters. The adapter-ligated DNA products were purified by paramagnetic beads. Adapter-ligated DNA was amplified in a linker-mediated PCR reaction (LM-PCR) for 11 cycles using a Phusion^TM^ DNA Polymerase and Illumina’s PE genomic DNA primer set and then purified by paramagnetic beads. Quality and quantity of the finished libraries were assessed using an Agilent HS DNA or DNA1000 chip (Agilent Technologies, Inc., Santa Clara, CA, USA) and a Qubit^®^ dsDNA HS Assay Kit (Invitrogen, Carlsbad, CA, USA), respectively. Libraries were standardized to 2 nM. Cluster generation was performed using standard Cluster Kits (v4) and the Illumina cBot. Single-end 100 bp sequencing was performed using standard SBS chemistry (v4) on an Illumina HiSeq2500 sequencer. Images were analyzed using the standard Illumina Pipeline, version 1.8.2.

### 2.11. Statistical Analysis

Viral quantities were compared among the EV-D68 strains using linear mixed-effect models, with fixed-effect terms for strain and random-effect terms for BEC donor. The Tukey adjustment was used to control the family-wise error rate for multiple comparisons between strains. Viral quantities were log-transformed for analysis. Statistical analysis was conducted using R 3.3.1 [23].

RNA sequencing data were analyzed in R using Bioconductor tools. Sequencing reads were aligned to the UCSC hg19 *Homo sapiens* genome, and gene counts were assigned using the Rsubread 1.22.3 package [24]. Gene annotation was performed using the org.Hs.eg.db 3.3.0 [25] and annotate 1.50.1 packages. Genes were filtered to include genes with >1 reads per million mapped reads in at least 2 libraries. The edgeR 3.14.0 package [26] was used to scale raw library sizes, estimate gene-wise dispersion using the empirical likelihood Bayes method, and fit quasi-likelihood negative binomial models with covariates for the BEC donor and virus strain to assess differential gene expression. Linear contrasts were used to compare gene expression patterns between the Fermon strain and the average expression of the other 4 EV-D68 strains. The false discovery rate was used to account for multiple testing.

## 3. Results

### 3.1. EV-D68 Binding and Replication in HeLa Cells

We first assessed binding and replication of recent EV-D68 isolates and Fermon using HeLa cells. This cell line was selected for its susceptibility to EV-D68 infection and genetic uniformity, allowing a comparison of viral binding and replication without the additional genetic variation observed in the primary cells from different donors. The results showed only minor differences among the recent clinical isolates of EV-D68 in either the binding or progeny yields (Figure 2A). In contrast, the Fermon strain demonstrated consistently greater binding (2–3 logs) and progeny yields (2–4 logs) at 2 and 24 h post infection (p.i.), respectively, when compared to the recent EV-D68 isolates. The ratio of 24 h over 2 h p.i. showed that the greater amount of virus progeny produced by Fermon was due to increased binding rather than enhanced viral RNA replication (Figure 2B).

### 3.2. Time Course of EV-D68 Binding and Replication in Primary BECs

To further examine the differences in binding and replication between the EV-D68 isolates, we used differentiated cultures of BEC, which are natural host cells for this virus. We first performed time-course experiments to identify the optimal time point p.i. for viral replication assessment. We used a recent EV-D68 clinical isolate from clade B1 (C7788) and Fermon. Both viruses had similar binding at 2 h; however, at 24 h, the C7788 strain demonstrated greater replication than Fermon (*p* < 0.05, Figure 3). There was no further increase in viral replication at 48 or 72 h p.i. in either strain, so we chose the 24 h p.i. time-point for further studies.

### 3.3. EV-D68 Binding and Replication in Primary BEC-ALI Cultures

Our preliminary experiments with two clinical isolates from the B1 clade (C7787 and C7788) and two variants of Fermon showed similar levels of binding between all four tested viruses but significantly higher (>one-log, *p* < 0.05) virus progeny yields and replication of clinical isolates compared to the Fermon variants in BEC-ALI cells (Appendix A). We next infected differentiated BEC-ALI cultures from five different donors with all available EV-D68 isolates and Fermon to examine the potential differences in virus binding and replication. In these experiments, there were no significant differences in the bound virus (2 h post-inoculation) between the recent isolates and Fermon (Figure 4A). In contrast, the recent isolates all had a 1.5–2 log greater viral yield and fold increase in EV RNA compared to the Fermon strain (*p* < 0.05) at 24 h p.i. Interestingly, the binding and replication levels were similar among the recent EV-D68 isolates that were obtained from asymptomatic infections vs. lower respiratory illnesses (Figure 4A,B).

### 3.4. Assessment of Host Immune Response to EV-D68 by RNA-Seq Analysis

We next tested for differences in the transcriptional responses elicited by EV-D68 infection in differentiated human primary BECs from five donors. We compared transcript abundances induced by the infection of BEC-ALI cultures with four EV-D68 isolates representing each phylogenetic clade (A: 0541a, B: C7731, B1: C7788, C: C2386, and Fermon), while controlling for differences among BEC donors. Differentially expressed (DE) genes were identified through pairwise comparisons of each isolate using a false discovery rate (FDR) of 0.2.

We first assessed genes that were differentially expressed after infection with clinical isolates from clades A, B, B1, and C compared to mock-infected control cells (Appendix A). We found a robust response in the infected BECs characterized by strong upregulation of antiviral and pro-inflammatory response genes such as OASL, DDX60, INFL1-3, CXCL10, and CXCL11 (Appendix A). There were no significant differences in gene expression among the clinical isolates (FDR < 0.2) (Appendix A). The Fermon strain induced a number of genes including heavy-metal-associated isoprenylated plant protein (HIPP), which is involved in the Hedgehog signaling pathway, and potassium two-pore domain channel subfamily K member 3 (KCNK3), which encodes a member of the two-pore potassium channel. In contrast to the recent clinical isolates, Fermon did not induce significant differences (FDR < 0.2) in the expression of antiviral and pro-inflammatory genes (Appendix A).

Given the similarity in the transcriptional response to the clinical isolates, we compared the transcriptional responses induced by Fermon to the average expression of genes induced by the four clinical isolates. This grouped analysis identified 90 DE genes (FDR = 0.2, Figure 5), and nearly all (88 of 90) of these genes were upregulated. Of the 35 genes (all upregulated) with an FDR < 0.05 and >2-fold change in expression (Table 2), most were related to host defense and antiviral responses (e.g., IFNL1, IFNL3, IFIT1, IFIT2, IFIT3, CXCL10, and CXCL11).

Given that Fermon replicated less efficiently than the recent isolates of EV-D68 in BEC, we next fit a model that included virus isolate, viral replication, and BEC donor. The results showed that isolate-specific differences in gene expression were attributable to differences in viral replication.

## 4. Discussion

The underlying mechanisms of the recent increase in frequency and severity of illnesses caused by EV-D68 are incompletely understood. This study utilized clinical isolates obtained over the past two decades as well as the prototype Fermon strain isolated in 1962 to assess whether changes in virus binding, replication, and/or the induction of immune response of recent EV-D68 isolates could provide potential explanations for these clinical observations. These studies identified distinct differences in viral replication and the epithelial immune response, as assessed by transcriptomics, between the prototype Fermon strain and the recent clinical isolates from clades A, B (including the B1 subclade), and C. Interestingly, no significant differences were identified in vitro among the recent EV-D68 clinical isolates despite the observed differences in associated disease severity, suggesting that individual factors are the main determinants of illness severity. The recent EV-D68 clinical isolates had similar binding and replication to one another but differed from the Fermon strain, which replicated better in HeLa cells but less well in BEC-ALI cultures. We further investigated the possible differences in host defense pathways by RNA-seq analysis and found an increase in some key host defense transcripts, such as IFNL1, IFNL3, CXCL10, and CXCL11, in response to infection with recent EV-D68 isolates compared to Fermon. Type-III interferons were also strongly induced by the contemporary EV-D68 isolates in PBE-ALI cells in a recent study [27]. These differences related closely to the degree of viral replication observed in these primary epithelial cell cultures. These findings suggest that an increased replication efficiency and enhanced inflammatory response induced by recent isolates in vitro could be responsible for the increased disease severity compared to the Fermon strain representing the past EV-D68 isolates from 1960s.

The recent isolates selected from children with either severe or asymptomatic infections had similar binding, replication, and immune responses in vitro. These findings suggest that host factors could be important in determining the severity of illness. In addition, changes in antigenicity leading to a loss of antibody protection [4] could have contributed to the recent severe outbreaks. The predilection for EV-D68 to more significantly impact young children suggests that the antibody response is likely important to the illness severity. Further, its cell specificity may have changed such that current strains can infect a broader range of cell types [28]. Indeed, some EV-D68 strains are capable of more effectively infecting cells through some unknown glycan-independent mechanism compared to Fermon [11]. The possible receptors that could allow EV-D68 to infect cells more effectively could be sulfated glycosaminoglycans, heparan sulfate proteoglycans, and/or ICAM-5 [9,10,29].

Our study had some strengths and limitations. One strength of our study was the use of differentiated cultures of human primary BEC, which provide a more physiologically relevant model system compared to monolayers of epithelial cells or continuous cell lines. Another important strength was the use of clinical virus isolates, representing all the major phylogenetic clades of EV-D68, that were cloned and produced by reverse genetics rather than ‘laboratory’ strains propagated in cell lines by infection. A limitation of the study was that the full passage history of the Fermon strain is unknown, and so it is possible that the original virus isolated in 1962 could be at least partially adapted to the RD or MRC-5 cell lines used for its propagation [30]. However, we found no significant differences in virus binding or replication between the Fermon variant received from Dr. Frank van Kuppeveld and the variant (Lab Fermon) cloned in our laboratory using the published genome sequence of the original clinical isolate from 1962 (Appendix A). Additionally, the primary human BECs were obtained from de-identified donors, so we do not know if they had underlying asthma or other respiratory conditions.

In conclusion, we identified differences in viral replication and epithelial cell responses to EV-D68 infection in vitro between the Fermon strain isolated in 1960s and the recent clinical isolates from clades A, B (including B1), and C that were obtained over the past two decades. These findings suggest that the enhanced viral replication and cellular inflammatory responses could be responsible for an increase in illness severity associated with recent isolates of EV-D68, and host factors are important in determining the severity of illness. Further studies are needed to fully understand the reasons underlying the EV-D68 outbreak in 2014.

## Figures and Tables

**Figure 1 viruses-15-01291-f001:**
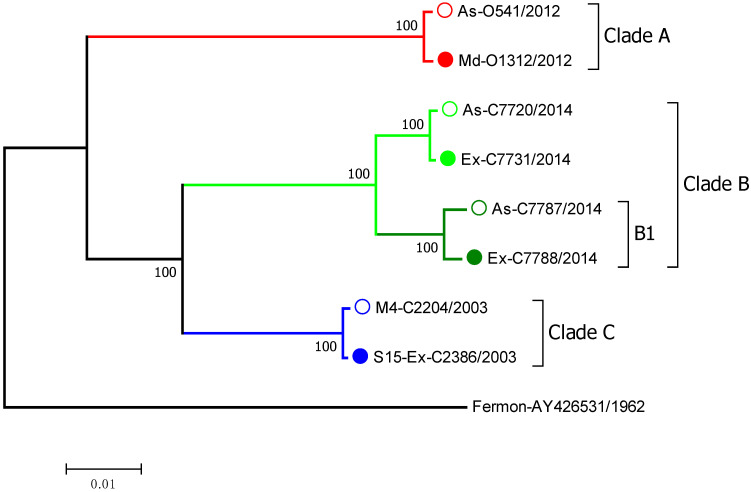
The phylogenetic tree of EV-D68 clinical isolates selected for cloning. Full-length genome sequences were analyzed. The numbers next to each tree branch represent bootstrap values showing the percentage of replicate trees that the viral strain clustered together. The branch lengths represent the evolutionary distance in units of base pair differences per viral sequence, with the scale bar denoting difference. Severity of symptoms designations: As—asymptomatic; Ex—exacerbation; M—mild; S—severe; numbers following the designations indicate symptom score where available. Filled and empty circles of the same color show symptomatic and asymptomatic infections, respectively.

**Figure 2 viruses-15-01291-f002:**
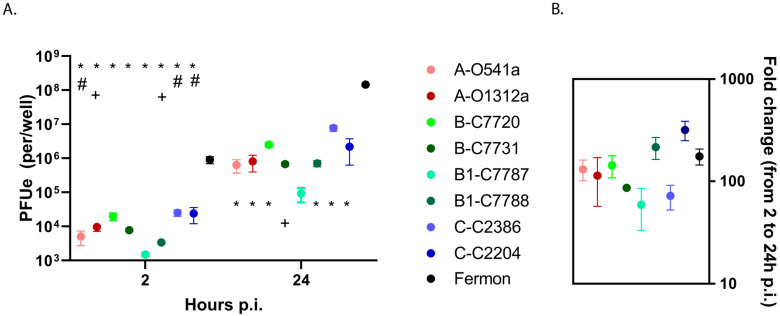
Viral binding and replication in HeLa cells. (**A**) Cell-associated input virus (2 h p.i.) and progeny yields (24 h p.i.) were determined by RT-qPCR and (**B**) viral replication was calculated and compared for each isolate. The first letters in isolate designations represent the phylogenetic clade, and the following letters and numbers indicate the clinical isolate ID. The * represents a difference (*p* < 0.05) in the clinical isolates vs. Fermon, + represents a difference (*p* < 0.05) in isolate 7787 vs. other isolates, and # represents a difference (*p* < 0.05) in isolates C2386, C2204, and C7720 vs. other isolates. Data are means and SEM from five independent experiments. The viruses are color-coded according to clade, with darker color indicating severe illness and lighter color indicating asymptomatic infection.

**Figure 3 viruses-15-01291-f003:**
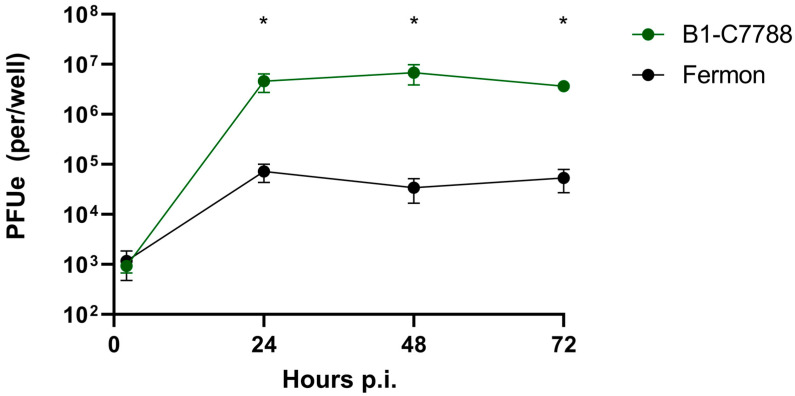
Time course of infection with isolate C7788 and Fermon in differentiated human primary BEC-ALI cultures. The * represents difference (*p* < 0.05) between Fermon and B1-7788 isolate. Data are means and SEM from five independent experiments.

**Figure 4 viruses-15-01291-f004:**
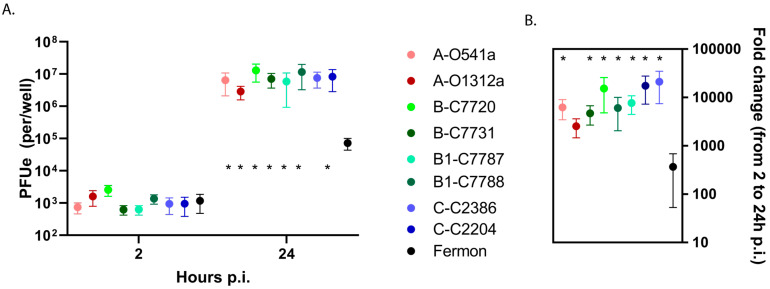
Viral binding and replication in differentiated human primary BEC-ALI. (**A**) Cell-associated input virus (2 h p.i.) and progeny yields (24 h p.i.) were determined by RT-qPCR and (**B**) viral replication was calculated and compared for each isolate. The first letters in isolate designations represent the phylogenetic clade, and the following letters and numbers indicate the clinical isolate ID. The * represents a difference (*p* < 0.05) between Fermon and the recent clinical isolates. Data are means and SEM from five independent experiments. The viruses are color-coded according to clade with darker color indicating severe illness and lighter color indicating asymptomatic infection.

**Figure 5 viruses-15-01291-f005:**
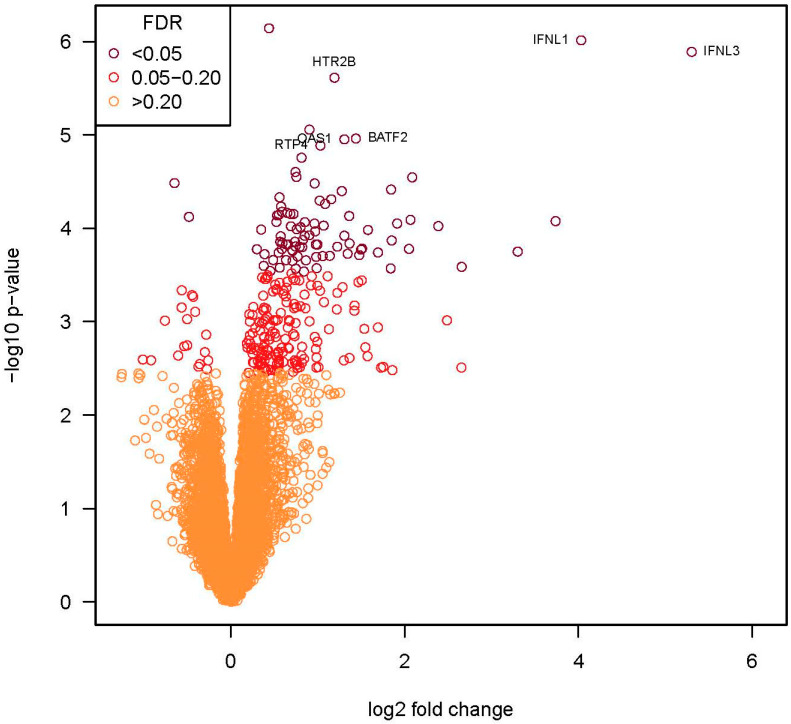
RNA-seq volcano plot identifying differentially expressed genes in BEC-ALI cultures after infection with recent EV-D68 isolates vs. Fermon. The *x*-axis represents the log_2_ values of the fold change observed for each mRNA transcript, and the *y*-axis represents the log_10_ of the *p*-values of the significance tests between replicates for each transcript. The orange circles represent genes that had an FDR of >0.2, circles in red are genes with an FDR between 0.05–0.2, and the circles in burgundy are genes with an FDR of <0.05. The labeled genes (n = 6) had an FDR of <0.03 and a fold change >2.

**Table 1 viruses-15-01291-t001:** Sequence accession numbers and restriction enzymes used for cDNA cloning.

Virus Strain	GenBank Accession Number	Restriction Enzyme (3′ End)	Restriction Enzyme (5′ End)
Lab Fermon	AY426531	Sall	XhoI
O541a	KX255369	MluI	Sall
O1312a	KX255358	MluI	SaII
C2386	KX255388	PstI	XhoI
C2204	KX255410	PstI	XhoI
C7720	KX255352	BstEII	XhoI
C7731	KX255354	BstEII	XhoI
C7787	KX255382	SpelI	XhoI
C7788	KX255413	SpelI	XhoI

**Table 2 viruses-15-01291-t002:** Differentially expressed genes * from comparison of recent EV-D68 isolates vs. Fermon.

Gene Name	Gene Symbol	Fold Change	*p*-Value	FDR *p*-Value
Interferon lambda 3	IFNL3	39.41	1 × 10^−6^	7 × 10^−3^
Interferon lambda 1	IFNL1	16.33	1 × 10^−6^	7 × 10^−3^
Interferon lambda 2	IFNL2	13.32	8 × 10^−5^	4 × 10^−2^
Interferon gamma-induced protein 10	CXCL10	9.85	2 × 10^−4^	4 × 10^−2^
Interferon gamma-induced protein 11	CXCL11	6.30	3 × 10^−4^	5 × 10^−2^
Interferon-induced protein tetratriopeptide repeats 2	IFIT2	5.23	1 × 10^−4^	4 × 10^−2^
2′-5′-oligoadenylate synthase	OASL	4.24	3 × 10^−5^	4 × 10^−2^
Z-DNA-binding protein 1	ZBP1	4.19	8 × 10^−5^	4 × 10^−2^
Interferon-induced protein tetratriopeptide repeats 1	IFIT1	4.14	2 × 10^−4^	4 × 10^−2^
Interferon-induced protein tetratriopeptide repeats 3	IFIT3	3.77	9 × 10^−5^	4 × 10^−2^
Radical SAM domain-containing 2	RSAD2	3.60	1 × 10^−4^	4 × 10^−2^
Cytidine/uridine monophosphate kinase 2	CMPK2	3.59	4 × 10^−5^	4 × 10^−2^
Interferon-induced GTP-binding protein Mx2	MX2	3.58	3 × 10^−4^	5 × 10^−2^
Interferon-induced protein AIM2	AIM2	3.23	2 × 10^−4^	4 × 10^−2^
Tumor necrosis factor receptor superfamily member 13B	TNFSF13B	2.98	1 × 10^−4^	4 × 10^−2^
Probable E3 ubiquitin-protein ligase HERC5	HERC5	2.85	2 × 10^−4^	4 × 10^−2^
Interferon-induced GTP-binding protein MX1	MX1	2.83	2 × 10^−4^	4 × 10^−2^
Epithelial stromal interaction 1	EPSTI1	2.78	2 × 10^−4^	4 × 10^−2^
Basic leucine zipper transcription factor, ATF-like 2	BATF2	2.71	1 × 10^−5^	2 × 10^−2^
2′-5′-oligoadenylate synthetase 2	OAS2	2.58	1 × 10^−4^	4 × 10^−2^
RIG-I (retinoic acid-inducible gene I)	DDX58	2.57	7 × 10^−5^	4 × 10^−2^
2′-5′-oligoadenylate synthetase 3	OAS3	2.54	2 × 10^−4^	4 × 10^−2^
2′-5′-oligoadenylate synthetase 1	OAS1	2.47	1 × 10^−5^	2 × 10^−2^
Probable E3 ubiquitin-protein ligase HERC6	HERC6	2.47	1 × 10^−4^	4 × 10^−2^
Ubiquitin-specific peptidase 18	USP18	2.42	4 × 10^−5^	4 × 10^−2^
XIAP-associated factor 1	XAF1	2.34	2 × 10^−4^	4 × 10^−2^
5-Hydroxytryptamine receptor 2B	HTR2B	2.28	2 × 10^−6^	9 × 10^−3^
T-cell activation RhoGTPase-activating protein	TAGAP	2.23	5 × 10^−5^	4 × 10^−2^
Interferon-induced transmembrane protein 1	IFITM1	2.20	2 × 10^−4^	4 × 10^−2^
Interferon-induced with helicase C domain 1	IFIH1	2.12	6 × 10^−5^	4 × 10^−2^
Interferon-induced guanylate-binding protein 1	GBP4	2.10	9 × 10^−5^	4 × 10^−2^
Probable ATP-dependent RNA helicase 60	DDX60	2.08	2 × 10^−4^	4 × 10^−2^
Receptor transporter protein 4	RTP4	2.04	1 × 10^−5^	2 × 10^−2^
Hematopoietic SH2 Domain	HSH2D	2.03	5 × 10^−5^	4 × 10^−2^

* FDR < 0.05 and fold change > 2.

## Data Availability

The RNA-seq data were deposited into the NCBI Gene Expression Omnibus (accession number GSE233630).

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
