# Peer review of "Recent Clinical Isolates of Enterovirus D68 Have Increased Replication and Induce Enhanced Epithelial Immune Response Compared to the Prototype Fermon Strain"

_viruses, 2023, doi:10.3390/v15061291_

Round 1

Reviewer 1 Report

This is a fine paper. Only a minor comment in terms of design of the study. It would have been interesting (in hindsight) to see the potential differences in transcriptional / innate profile between 'severe' and 'mild/asymptomatic' variants within each clade (if they exist), which as far as I can see has not been assessed.

Author Response

We appreciate your input in reviewing our manuscript and the helpful comment. We have not assessed differences in the innate responses induced by isolates that caused “severe” vs. “'mild/asymptomatic” symptoms within each clade. We selected only one representative virus isolate from each clade. They included one isolate from asymptomatic infection (O541) and 3 isolates associated with asthma exacerbations (C7731, C7788 and C2386). We didn’t find statistically significant differences in gene expression between the selected isolates. We added additional supplementary figure (Figure S2) to show these findings. We didn’t find significant differences in virus binding or replication in primary airway epithelial cells in vitro between the isolates based on severity of symptoms. Taken together, these findings suggest that host factors are likely the most important in determining severity of illness (page 12, lines 351-353, 367-368).

Reviewer 2 Report

Dear authors,

I have no comments on your manuscript. The work is interesting and important for establishing changes in the epidemic significance of EV-D68.

Author Response

We appreciate your positive comments about our paper.

Reviewer 3 Report

Devries et al. studied the pathogenesis of EV-D68 in a biologically relevant model by using primary bronchial epithelial cells from 5 de-identified donors grown in air-liquid interface. Comparison of viral binding and replication was done not only between HeLa cells and primary BECs, but also between recent clinical isolates and the prototype strain Fermon. In this study, cDNA infectious clones were generated from 8 clinical isolates, which belong to the major clades of EV-D68. The strains were classified nicely into those causing mild, severe and asymptomatic diseases. The study showed that recent clinical isolates replicate more efficiently and induce more robust immune response in primary BECs than in HeLa cells than Fermon cells.

I have several questions, comments and suggestions that can hopefully improve this manuscript:

1.     What cells are WisL cells? 

2.     It is unclear why primary BECs were chosen instead of primary nasal and/or small-airway epithelial cells. The authors briefly mentioned that BEC are natural host cells for this virus. However, this does not fully explain the virus optimal growth at 33 C (i.e. more optimally growing in nose than in bronchi, thus nasal epithelial cells seem to be more natural host cells than the BECs). See also comments below (#3).

3.     In Materials & Methods, it is not described at which temperature the primary BECs were maintained. Some studies have described maintenance of their primary BECs at 37 C, since the airway mucosal temperature of human segmented bronchi is ~37 C. The temperature of the upper respiratory tract, such as the nose, is much lower, ranging between 30 and 34 C. In this study, the inoculation was performed at 34 C for 2 h. The authors should also provide an explanation why they have decided to perform the inoculation in a less optimal environment.

4.     It is unclear what the multiplicity of infection in BEC ALI culture was. Titre of inoculum was also not provided, so it is impossible to judge if the cells were indeed inoculated with the same amounts of viruses (105 PFUe/well).

5.     Why did the authors use a different inoculum amount for HeLa cell infection (2.5e6 PFUe/well) than BEC (105 PFUe/well)? Is the ratio of cells and viruses in HeLa equivalent/comparable to that in BEC?

6.     In regards to comments #4, it is difficult to judge if differences at 2 hpi (Fig 1) are therefore due to binding rather than differences in inoculum. 

7.     The authors concluded that the greater amount of Fermon virus progeny in Fig 1 is due to increased binding. However, the authors did not perform a binding study to show that this is indeed the case. It is still possible that certain strains just replicate more efficiently in a certain cell line, resulting in higher detectable level of RNA intracellularly, thus not necessarily having better binding.

8.     It is unclear why the clinical isolate B1 (C7788) was chosen as a representative for the time course experiment.

9.     It will be interesting to also visualise the data of differentially expressed genes in BEC ALI cultures after inoculation with different clinical isolates (in other words, instead of comparing clinical isolates with Fermon, show the comparison of the clinical isolates from clades A, B and C). The authors briefly mentioned that the transcriptional response to the clinical isolates were similar, but I think it is important to show this and highlight this message more strongly, since it can have a clinical relevance and an importance in pathogenesis study.

10.  Cytopathic effect, ciliary movement and mucus production following the infection are not described in BEC ALI culture. Are there differences in CPE, ciliary movement and mucus production following infection of different clinical isolates (and Fermon)?

11.  Lines 381-383: for development of severe disease is often viral interference or escape from immune responses or prolonged inflammatory immune responses. The authors have only tested up to 24 hpi. It cannot be concluded that the “increased” inflammatory responses observed at that time point may contribute to severe diseases, especially because the authors have included strains that have caused mild and severe diseases, which apparently showed similar transcriptional responses in BEC ALI cultures.

Author Response

  1. What cells are WisL cells?

These are human embryonic lung fibroblasts that are similar to commercially available WI-38 and MRC-5 cells (page 4, lines 125-126).

  1. It is unclear why primary BECs were chosen instead of primary nasal and/or small-airway epithelial cells. The authors briefly mentioned that BEC are natural host cells for this virus. However, this does not fully explain the virus optimal growth at 33 C (i.e.more optimally growing in nose than in bronchi, thus nasal epithelial cells seem to be more natural host cells than the BECs). See also comments below (#3).

Since most of the isolates that caused severe illnesses were associated with asthma exacerbations, we thought that bronchial epithelial cells would be a more relevant model than nasal epithelial cells for the in vitro studies. We have not compared viral replication at 33°C vs. 37°C but found high levels of replication at 34°C.

  1. In Materials & Methods, it is not described at which temperature the primary BECs were maintained. Some studies have described maintenance of their primary BECs at 37 C, since the airway mucosal temperature of human segmented bronchi is ~37 C. The temperature of the upper respiratory tract, such as the nose, is much lower, ranging between 30 and 34 C. In this study, the inoculation was performed at 34 C for 2 h. The authors should also provide an explanation why they have decided to perform the inoculation in a less optimal environment.

We grew BEC ALI cultures at 37°C. We now added this information to section 2.7. of the Materials and Methods (page 5, lines 171-174). As we explain in our answer to the previous question, we have not compared viral replication at 33°C vs. 37°C but found high levels of replication at 34°C. Either similar or even slightly higher levels of EV-D68 replication were found at 33° C vs. 37° C in a recent study utilizing similar PBE-ALI cell cultures [PMID: 34196272].

  1. It is unclear what the multiplicity of infection in BEC ALI culture was. Titre of inoculum was also not provided, so it is impossible to judge if the cells were indeed inoculated with the same amounts of viruses (105PFUe/well).

The BEC-ALI cultures were seeded at ~100K per well, grown in transwell inserts in 24-well plates until full differentiation and infected with the same dose of each virus per well (105 PFUe diluted in 50 ul of PC-ALI medium / well). The total number of cells in those transwells at the time of infection is typically 400-500K. It’s difficult to accurately assess the MOI in differentiated pseudostratified epithelial cell cultures but we estimate it to be an MOI of 0.5-1 PFU per cell at the dose we used.

  1. Why did the authors use a different inoculum amount for HeLa cell infection (2.5e6PFUe/well) than BEC (105 PFUe/well)? Is the ratio of cells and viruses in HeLa equivalent/comparable to that in BEC?

We estimate about 500K HeLa cells per well in a 12-well plate at the time of infection so the MOI used for HeLa infection (5 PFU/cell) was higher than that in PBE-ALI cells (0.5-1 PFU/cell). We used a higher MOI in HeLa cells because viral replication is less in these cells compared to PBE ALI cells. We compared the virus isolates within each cell type, therefore, using different virus dose / MOI didn’t affect the results and conclusions.

  1. In regards to comments #4, it is difficult to judge if differences at 2 hpi (Fig 1) are therefore due to binding rather than differences in inoculum. 

We didn’t find statistically significant differences in binding to BECs between the isolates tested. Binding of the recent clinical isolates to HeLa cells was lower than that of Fermon strain. There were also some small differences in binding to HeLa cells found between some of the isolates.

  1. The authors concluded that the greater amount of Fermon virus progeny in Fig 1 is due to increased binding. However, the authors did not perform a binding study to show that this is indeed the case. It is still possible that certain strains just replicate more efficiently in a certain cell line, resulting in higher detectable level of RNA intracellularly, thus not necessarily having better binding.

The differences that we found at 2h. p.i. in HeLa cells reflect differences in binding / cell entry step. We thoroughly rinsed the cells 3 times with PBS to remove unbound virus. Viral replication starts at later time points after infection. We estimated viral replication as a fold increase in viral RNA load from 2h (input virus) to 24 h (progeny virus) post infection.

  1. It is unclear why the clinical isolate B1 (C7788) was chosen as a representative for the time course experiment.

This isolate was chosen because it was one of the recent viruses circulating during the 2014 outbreak that belonged to the “newly emerged” clade B1. 

  1. It will be interesting to also visualise the data of differentially expressed genes in BEC ALI cultures after inoculation with different clinical isolates (in other words, instead of comparing clinical isolates with Fermon, show the comparison of the clinical isolates from clades A, B and C). The authors briefly mentioned that the transcriptional response to the clinical isolates were similar, but I think it is important to show this and highlight this message more strongly, since it can have a clinical relevance and an importance in pathogenesis study.

As the reviewer correctly noted, we didn’t find any statistically significant differences in gene expression between the selected isolates. We now added additional supplementary figure (Figure S2) to show these findings. Please also see the response to reviewer 1 comment.

  1. Cytopathic effect, ciliary movement and mucus production following the infection are not described in BEC ALI culture. Are there differences in CPE, ciliary movement and mucus production following infection of different clinical isolates (and Fermon)?

We did not specifically measure ciliary beating or mucus secretion, however, there were no differences noted by light microscopy in severity of CPE or ciliary movement after infection with different isolates. No clear CPE was visible by microscopy similarly to infection of PBE-ALI cultures with other respiratory viruses (e.g. RV).

  1. Lines 381-383: for development of severe disease is often viral interference or escape from immune responses or prolonged inflammatory immune responses. The authors have only tested up to 24 hpi. It cannot be concluded that the “increased” inflammatory responses observed at that time point may contribute to severe diseases, especially because the authors have included strains that have caused mild and severe diseases, which apparently showed similar transcriptional responses in BEC ALI cultures.

We agree with the reviewer that both escape from immune responses and prolonged inflammatory immune responses may contribute to disease severity. However, increased viral replication and inflammatory responses observed at 24 h in vitro (compared to Fermon) could also contribute to disease severity of recently emerged D68 isolates in vivo. The lack of differences in vitro between genetically close isolates associated with either severe illnesses or mild / asymptomatic infections suggests that host factors are likely the most important in determining severity of illness (page 12, lines 351-353, 367-368). We also streamlined these results in the revised Abstract (page 1, lines 23-25, 32).

Round 2

Reviewer 3 Report

The authors have improved the manuscript and addressed my questions sufficiently. I have no further comments or questions.